# Helping Across Boundaries: Collectivism and Hierarchy in the Ultra-Orthodox Context

**DOI:** 10.3390/bs15040520

**Published:** 2025-04-13

**Authors:** Chananel Goldfinger, Shomi Shahar-Rosenblum, Itschak Trachtengot, Nechumi Malovicki-Yaffe

**Affiliations:** 1School of Social Work, University of Haifa, Haifa 3103301, Israel; cgoldfin@campus.haifa.ac.il; 2Baruch Ivcher School of Psychology, Social Psychology, Reichman University, Herzliya 4610101, Israel; shimonmeirchay.shah@post.runi.ac.il; 3Psychology Department, Faculty of Social Sciences, The Hebrew University of Jerusalem, Jerusalem 9190500, Israel; itschak.trachtingot@mail.huji.ac.il; 4Public Policy Department, School of Social Sciences, Tel-Aviv University, Tel Aviv 6997801, Israel

**Keywords:** collectivism, individualism, Ultra-Orthodox, HVIC, prosocial behavior, in-group preference

## Abstract

Understanding the role of collectivism in shaping prosocial behavior is critical for advancing theories of social cooperation and group dynamics. This study provides the first empirical examination of collectivistic orientation within the Ultra-Orthodox (Haredi) community using the Horizontal and Vertical Individualism-Collectivism (HVIC) framework. Data from 702 participants revealed a predominant collectivist orientation, with a particularly strong emphasis on balanced collectivism. The study further explored how collectivist and individualist tendencies predict helping behaviors toward in-group and out-group members. Results indicate that conservatism positively predicts in-group prosocial behavior but negatively predicts out-group assistance, whereas balanced collectivism and individualism are associated with increased out-group helping. The strongest predictor of out-group assistance was an individual’s inherent disposition to help, suggesting that prosocial behavior extends beyond purely communal expectations and positions these individuals as natural agents of community change. This insight offers a perspective on how personal characteristics may contribute to community renewal. Our study contributes to cross-cultural research on collectivism and prosocial behavior by emphasizing the role of power orientation and resource allocation in shaping altruistic tendencies, while demonstrating that vertical orientations tend to reinforce in-group preferences.

## 1. Introduction

Cultural variations among societies are often examined through the individualism-collectivism dimension, a foundational framework introduced in [49] ([49]) cultural values model. This framework explores how societies define self-boundaries, shape personal identity, and balance individual needs with group obligations ([78]; [61]; [113]). Later refinements to this model introduced power dynamics, distinguishing between horizontal (egalitarian) and vertical (hierarchical) dimensions of individualism and collectivism ([29]; [102]; [103]). These conceptual advancements allow for a more nuanced understanding of cultural variations in social behavior, group cohesion, and prosocial tendencies.

The present study applies these theoretical frameworks to Israel’s Ultra-Orthodox (Haredi) community. This community has attracted increasing academic interest due to its demographic growth, social insularity, and strong communal structures. Despite constituting a relatively small proportion of the overall population, the Haredi community wields significant political influence due to Israel’s coalition-based system of governance, which often grants minority groups substantial bargaining power in shaping national policies. ([74]). While scholars have frequently described the Haredi community as collectivistic ([4]; [32]; [101]), few empirical studies have systematically examined its collectivist orientation and its implications for social behavior.

Building on this foundation, this study aims to empirically assess the collectivist nature of the Haredi community through the Horizontal–Vertical Individualism–Collectivism (HVIC) framework. Additionally, it seeks to analyze patterns of prosocial behavior within and beyond community boundaries to determine whether helping behaviors stem primarily from collectivist values or are influenced by other underlying factors.

While collectivist communities typically emphasize in-group solidarity and mutual aid, the extent to which these behaviors extend beyond group boundaries remains an open question. Previous research suggests that vertical collectivist cultures often prioritize group hierarchy and in-group obligations over out-group assistance ([113]; [66]). Conversely, horizontal collectivist orientations may foster broader prosocial tendencies, encouraging acts of altruism across social divisions ([13]). By exploring these dynamics, the present study seeks to contribute to the broader discourse on collectivism, power orientation, and prosocial behavior.

### 1.1. Individualism-Collectivism Dimension

The systematic categorization of global cultures began with [49] ([49]) introduction of the individualism–collectivism dimension, providing researchers with a clear framework for cultural classification. In collectivist societies, individuals develop their identity through their social context, perceiving themselves as integral components of their group rather than independent entities ([95]; [117]). These societies prioritize cooperation and group cohesion, fostering social structures that reinforce interdependence and shared responsibilities ([77]; [18]). Individual identity emerges primarily through social relationships and group roles ([114]).

Members of collectivist societies maintain clear distinctions between in-group and out-group members ([122]), while emphasizing interpersonal relationships and social harmony over individual achievements ([51]; [54]). This collective orientation manifests prominently in family structure, where extended family units feature close connections across multiple generations and family branches ([68]; [112]).

In contrast, individualistic societies emphasize personal autonomy and individual distinctiveness. Their members function as independent entities with clear boundaries, forming social connections based on flexibility and personal choice ([114]; [117]). These societies prioritize privacy and personal autonomy as central values ([50]), typically organizing around small nuclear family units consisting primarily of parents and their children ([40]; [116]). These fundamental cultural differences extend beyond family structure, influencing various social and economic domains, from educational practices to economic decision-making ([1]; [26]; [69]).

While the individualism–collectivism framework has been valuable in cross-cultural research, it does not fully account for the nuanced variations within societies that exhibit both collective and individual orientations.

The traditional individualism–collectivism dichotomy, while valuable, does not fully account for variations within societies that share similar collective or individual orientations. Cultural analysis becomes more nuanced when incorporating power dynamics, revealing how hierarchical versus egalitarian structures shape social patterns beyond the basic individualism–collectivism distinction. These power relations manifest uniquely across different societies: in collectivist cultures, where identity is embedded within the group, and in individualistic societies, where independent identity takes precedence ([29]).

The contrast between hierarchical and egalitarian structures within individualistic and collectivist societies illustrates these complexities ([103]; [13]). Despite their shared individualistic orientation, American culture emphasizes competition and hierarchical achievement, whereas Swedish society prioritizes equality and individual self-reliance ([105]). A similar distinction is evident among collectivist societies, as seen in the hierarchical organization of Japan, which contrasts with the egalitarian ethos of the Israeli kibbutz movement ([105]). These differences highlight how power relations transcend the basic individualism–collectivism framework and influence broader social structures.

To integrate these power dynamics into cultural analysis, researchers developed a framework that examines both vertical (hierarchical) and horizontal (egalitarian) dimensions alongside individualistic and collectivistic orientations. This refined model captures the ways in which societies balance individual and group needs while simultaneously managing power relationships. The horizontal–vertical individualism–collectivism (HVIC) framework accounts for both a society’s orientation toward individual or collective identity and its approach to social hierarchy ([29]).

Horizontal individualism (HI) reflects a preference for personal autonomy while rejecting status-based differentiation, promoting equality among individuals. This orientation is characterized by statements such as “*I’d rather depend on myself than others*” and “*My personal identity, independent of others, is very important to me*”. In contrast, vertical individualism (VI) is similarly rooted in personal autonomy but is embedded within a hierarchical and competitive social structure in which individuals strive for status and achievement. This orientation is reflected in statements such as “*Winning is everything*” and “*It is important that I do my job better than others*”.

While individualism can be either horizontal or vertical, collectivism is similarly divided along these two dimensions. Horizontal collectivism (HC) emphasizes equality within the group, fostering shared goals and mutual interdependence while rejecting social stratification. This orientation is captured in statements such as “*To me, pleasure is spending time with others*” and “*The well-being of my coworkers is important to me*”. ([13]) In contrast, vertical collectivism (VC) views individuals as part of a hierarchical group working toward collective goals, where members willingly sacrifice personal interests and submit to authority for the benefit of the group ([103]; [115]). This orientation is reflected in statements such as “*It is my duty to take care of my family, even when I have to sacrifice what I want*” and “*It is important to me that I respect the decisions made by my group*”.

Originally, a 32-item questionnaire was developed to measure these four dimensions, as proposed by [114] ([114]). However, [105] ([105]) identified methodological limitations in the original scale and introduced a more concise 14-item version with improved validity. The present study employs this refined 14-item scale to examine Israel’s Haredi (Ultra-Orthodox) society, investigating both its collectivist orientation and its influence on intergroup relationships. This research constitutes the first systematic application of the HVIC framework to the Haredi community, offering a novel empirical perspective on how hierarchical and egalitarian structures shape prosocial behavior within and beyond the community.

### 1.2. Collectivism-Individualism and Group Preference

A central issue in studying differences between collectivist and individualistic cultures involves their approaches to in-group preference and attitudes toward out-groups. Cultural values theory suggests that individuals from collectivist cultures should demonstrate stronger preference for their in-group ([66]). However, empirical research presents a more complex picture, with some studies finding no support for stronger in-group favoritism in collectivist societies ([48]; [123]).

[66] ([66]) found contrasting patterns in group interactions across different societies: individualistic societies exhibited more positive in-group interactions, while collectivist cultures showed stronger hostility toward out-groups. The researchers attribute this pattern to the higher social mobility in individualistic societies, where individuals actively invest in positive relationships due to their ability to choose and change social connections. In contrast, collectivist societies, characterized by more fixed relationships, tend to emphasize negative distinctions toward out-groups.

Different emotional expression patterns also emerge across these cultural orientations. [80] ([80]) observed that individualistic cultures tend to express more negative emotions toward in-groups and fewer toward out-groups, while collectivist cultures display the opposite pattern—a difference attributed to the emphasis on maintaining group harmony in collectivist societies ([124]).

Research on group norms provides additional insights into these dynamics. [41] ([41]) found that collectivist norms correlate with stronger in-group preference compared to individualistic norms, with this difference becoming more pronounced under conditions of existential threat. Supporting these findings, a meta-analysis by [33] ([33]) revealed that when examining real social groups, collectivist societies demonstrated stronger in-group preference compared to individualistic societies.

### 1.3. Prosocial Behavior Between Groups in Collectivist and Individualistic Societies

Prosocial behavior is defined as voluntary actions intended to benefit others or promote positive relationships with others. It encompasses a wide range of behaviors, from everyday assistance to significant contributions of resources and effort ([27]; [16]). The relationship between cultural values of collectivism and prosocial behavior has been studied in various contexts, with research indicating complexity in the relationships between these variables ([83]).

Studies examining patterns of prosocial behavior have found that collectivist societies show greater willingness to provide assistance ([98]; [106]; [120]), including toward out-group members ([81]; [83]). According to [120] ([120]), these findings might also be explained by society’s level of religiosity. However, [65] ([65]) found no significant correlation between the collectivism–individualism dimension and willingness to help. Research examining the relationship between national culture and prosocial behaviors across 66 countries found that, contrary to the common assumption that collectivist societies would be more generous, individualistic societies actually demonstrated more prosocial behaviors ([67]). Yet, a review on the relationship between culture and prosocial behavior ([30]) indicates that collectivist cultures showed a higher tendency toward prosocial behavior, particularly toward in-group members, while individualistic cultures showed less distinction between helping in-group versus out-group members. According to [13] ([13]), only horizontal collectivism was found to be associated with a higher tendency toward prosocial behavior. This contrasts with findings by [21] ([21]), who identified vertical collectivism as one of the strongest predictors of prosocial behavior during the pandemic.

Another significant aspect in understanding cross-cultural helping behavior involves the embeddedness-autonomy dimension. [96] ([96]) developed this dimension as part of his cultural values model to examine the relationship between individual and group across different cultures. Research has demonstrated substantial overlap between the embeddedness-autonomy measure and the collectivism–individualism dimension, with studies showing high correlations between these constructs ([42]; [97]). In a key study examining the relationship between helping behavior and cultural embeddedness, [62] ([62]) found that higher levels of embeddedness correlated with reduced willingness to help strangers.

Self-construal represents another important framework for understanding cultural differences in helping behavior. [78] ([78], [79]) distinguish between independent self-construal, characteristic of individualistic cultures, and interdependent self-construal, prevalent in collectivist cultures ([20]). Research examining the relationship between self-construal and prosocial behavior has yielded mixed findings. [25] ([25]) found that individuals with independent self-construal showed similar levels of helping behavior toward both in-group and out-group members, while those with interdependent self-construal demonstrated stronger preference for helping in-group members. However, contradicting these findings, [5] ([5]) suggest that societies characterized by interdependence actually show higher levels of helping behavior toward both in-group and out-group members.

Notably, while the relationship between cultural values and prosocial behavior has been extensively studied, the specific impact of horizontal and vertical dimensions of individualism–collectivism (HVIC) on in-group versus out-group prosocial behavior remains unexplored. This represents a significant gap in our understanding of how power orientation within societies might influence helping patterns across group boundaries.

### 1.4. Religiosity and Group-Boundaries in Prosocial Behavior

The relationship between religiosity and prosocial behavior has been extensively explored, particularly with regard to in-group versus out-group helping. A recent meta-analysis found a small but robust positive association between religiosity and prosociality, with stronger effects for self-reported behavior than for behavioral measures. Notably, religious individuals were more inclined to help in-group members, although this tendency was moderate in size ([60]). This pattern is supported by studies reporting selective prosociality among religious populations ([9]; [90]; [107]). These studies indicate a tendency to favor co-religionists and exhibit reduced support or more negative attitudes toward out-groups such as atheists or members of other religions.

In contrast, several studies suggest that religiosity can promote generalized prosociality, particularly under certain cognitive or contextual conditions. Experimental findings show that religious priming enhances generosity toward both in-group and out-group members ([15]; [84]). Similar effects were observed among Muslim minorities, where priming increased donations primarily toward non-Muslim recipients ([82]). A field experiment in China found that Christians and Buddhists showed no clear in-group preference, whereas Muslim participants favored their own group ([118]). Additional support for the link between religiosity and broader altruistic orientation comes from a study among Jewish populations in Israel, which found that higher levels of religiosity were positively associated with general altruistic tendencies ([23]). A systematic review identified mechanisms through which religiosity shapes giving to both in-group and out-group recipients, highlighting cross-religious variation ([121]).

Taken together, the literature suggests a dual dynamic: while religiosity often enhances prosocial behavior toward the in-group, it may also motivate helping across group boundaries when moral reflection, minority status, or religious mission are salient. This distinction is particularly relevant in collectivist religious communities, such as the Ultra-Orthodox population studied here.

### 1.5. The Israeli Ultra-Orthodox (Haredi) Community

The Ultra-Orthodox (Haredi) community represents a distinct group within the broader Jewish population, characterized by strict adherence to religious law ([35]). This community places Jewish religious observance, as interpreted by its leaders and members, at the center of its existence ([14]; [89]).

As of 2024, Israel’s Ultra-Orthodox community numbers approximately 1,392,470 people, comprising 13.9% of Israel’s total population, with a 4.2% annual growth rate ([70]). The average Ultra-Orthodox family includes 5.15 members ([88]). Due to large family sizes, limited secular education, and low workforce participation, the Ultra-Orthodox community remains among Israel’s economically disadvantaged populations ([59]; [76]; [56]).

Charitable giving and acts of kindness represent a distinctive characteristic of the Ultra-Orthodox (Haredi) community ([99]; [75]). Research demonstrates that Haredi community members lead in volunteer participation across Israel’s charitable organizations ([108]). In a study of national civic service, [71] ([71]) found that most young Haredi men preferred to volunteer within their own community, citing a sense of security and belonging associated with bonding social capital. However, a smaller number opted for extra-community service, motivated by bridging social capital and a desire for broader integration into Israeli society. A particularly noteworthy finding concerns organ donations, where the majority of donors come from religious and Ultra-Orthodox sectors ([11], [12]; [23]). While most Haredi donors expressed a preference for Jewish recipients ([11], [12]), they showed relative indifference to the recipient’s level of religious observance ([12]).

This pattern of extensive charitable activity exists within a broader context of cultural distinctiveness. The Haredi community maintains an “enclave culture” within Israeli society ([45]), deliberately minimizing contact with secular and modern influences ([43], [44]; [72]; [110]). This separation shapes their social and institutional practices, as well as their vision of proper Jewish life ([44]; [10]). The community’s distinct ideological positions create unique socialization processes ([52]; [56]), which both reinforce internal solidarity and shape patterns of interaction with the broader society ([111]).

Historically, the Haredi community in Israel has been categorized primarily along ethnic and geographical lines: Ashkenazi Jews of European origin—further divided into Hasidic and Lithuanian (non-Hasidic) groups—and Sephardic Jews from Muslim countries ([14]; [64]). These distinctions emerged from diverse religious practices and institutional developments shaped in the diasporic experience.

In recent decades, however, the Ultra-Orthodox community has undergone significant social and ideological transformations. Its social margins have expanded, with new subgroups and ideological currents emerging ([63]; [47]; [2]). Increasing numbers of Haredi individuals have entered higher education, the workforce, and public life more broadly ([56]; [57]), while a nascent middle-class lifestyle has begun to develop within parts of the community, reflecting greater openness to modernization and societal integration ([109]).

These developments have rendered traditional ethnic and sectarian categories insufficient for capturing the current complexity of the community. The boundaries that once defined Haredi identity—based on origin or affiliation—are increasingly challenged by new patterns of lifestyle, ideology, and degrees of engagement with secular society. As such, there is a growing need for alternative analytical frameworks that can better account for internal differentiation and evolving boundary dynamics.

In response to this complexity, [76] ([76]) propose a more nuanced analytical model based on two intersecting axes: halakhic conservatism and sociological conservatism. The halakhic axis measures the stringency of religious law observance, while the sociological axis captures the extent of conformity to communal norms and the degree of separation from secular influences, as seen in lifestyle choices such as dress, media exposure, and cultural consumption. This dual-axis framework enables a richer understanding of the community, one that captures both religious commitment and strategies of boundary maintenance vis à vis the secular world.

However, despite these internal divisions, research consistently identifies strong collectivist patterns that transcend these subgroup boundaries, characterizing the Haredi community as a whole.

### 1.6. Collectivism in the Ultra-Orthodox Community

Research consistently characterizes the Ultra-Orthodox (Haredi) community as embodying collectivist cultural values ([4]; [32]; [101]). This collectivism manifests through multiple dimensions: communal responsibility for supporting individual members ([37]), strong mutual commitment between community members ([39]), adherence to social institutions that define individual identity ([35]), and the prioritization of community welfare over individual interests ([38]).

These collectivist tendencies shape complex social dynamics, particularly in the community’s response to norm violations. The community employs social sanctions against those who deviate from established norms ([58]; [73]; [63]; [53]), leading individuals to sometimes conceal problems and challenges to avoid stigmatization ([34]; [39]). While this system subordinates individualistic values such as personal rights, self-actualization, and privacy, it simultaneously creates robust intra-community support networks that provide significant social safety nets for community members ([7]).

The collectivist framework has proven valuable for understanding various aspects of Haredi life. Researchers have applied this perspective across multiple domains: mental health treatment ([24]; [34]; [85]) social work and welfare ([38]; [37]), vocational rehabilitation ([93]), family dynamics ([32]), education ([91]; [58]), problem-solving approaches ([4]), work values ([101]), life domain priorities ([100]), and responses to social rejection ([119]).

Despite the widespread theoretical characterization of the Haredi community as collectivistic, this fundamental assumption lacks direct empirical validation. While numerous studies cite the community’s collectivistic nature, systematic empirical research examining collectivism within the Haredi population is notably absent. The few existing studies have focused only on the broader religious Jewish population: [94] ([94]) found higher collectivism levels among religious students compared to their secular counterparts, while [19] ([19]) documented moderate collectivism levels among religious Jewish teachers—higher than secular Jews but lower than Arabs and Druze. This gap extends beyond the specific horizontal–vertical individualism–collectivism (HVIC) framework to include any empirical measurement of collectivistic tendencies within the Haredi community itself.

### 1.7. Study Goal

The present study had two primary objectives: (1) to investigate how HVIC dimensions influence in-group versus out-group prosocial behavior—a relationship not yet explored in the literature, and (2) to examine the applicability of the HVIC framework within the Ultra-Orthodox population, assessing how these orientations shape willingness to assist both community members and individuals outside the community.

Based on previous research, we formulated several hypotheses. Regarding cultural orientation, we hypothesized that (H1a) the Ultra-Orthodox community would exhibit stronger collectivist tendencies than individualist ones. Additionally, we expected that (H1b) the community would display a hierarchical orientation, with vertical dimensions scoring higher than horizontal ones, even within individualistic values.

In relation to helping behavior, we hypothesized that (H2a) willingness to help in-group members would be significantly higher than willingness to help out-group members. Furthermore, we predicted that (H2b) in-group helping behavior would primarily be driven by collectivist values (both horizontal and vertical), whereas out-group helping behavior would be associated with individualistic values. Lastly, we expected that vertical orientations (both collectivistic and individualistic) would be linked to a stronger preference for in-group helping behavior.

## 2. Materials and Methods

### 2.1. Participants and Sampling Procedures

From an initial sample of 843 participants, 702 met the inclusion criteria for final analysis. Data cleaning followed a systematic process that excluded the following: participants from the secular sector (n = 19), those outside the age range of 18–75 years (n = 73), respondents with anomalous response times exceeding 2.5 standard deviations from the mean (n = 14), and individuals residing abroad (n = 35).

The final sample (N = 702) consisted of 64% males and 36% females, with ages ranging from 18 to 75 years (M = 30.98, SD = 11.81). Participants represented various Haredi subgroups: Lithuanian (33.3%), Hasidic (23.4%), Sephardic (23.2%), Chabad (6.4%), Hardali (5.4%), and other affiliations (8.3%).

Using [76] ([76]) framework, participants’ religious conservatism was assessed through two measures: halakhic meticulousness and adherence to Haredi sociological norms. The combined hardiness scale categorized participants into four groups: Modern (14.1%), Modern-leaning (33.0%), Classical (33.9%), and Conservative (18.9%).

### 2.2. Measures

All measures employed 6-point Likert-type scales ranging from 1 (not at all) to 6 (to a great extent) unless otherwise specified. The individualism-collectivism dimensions were assessed using [105] ([105]) scale, which measured **balanced collectivism** (α = 0.499, M = 4.22, SD = 0.89), **balanced individualism** (α = 0.665, M = 2.80, SD = 1.17), **vertical collectivism** (α = 0.589, M = 4.19, SD = 0.99), and **vertical individualism** (α = 0.622, M = 3.34, SD = 1.21).

**Intergroup helping behavior** was measured using parallel scales adapted from [55] ([55]), assessing participants’ willingness to help both in-group and out-group members. The scale included three parallel items for each group, examining willingness to sign a petition supporting a social cause, donate money to a charity organization, and donate bone marrow to a patient. Each scenario was presented twice, once referring to Haredi recipients (in-group) and once to secular recipients (out-group).

**Hardiness Scale**—Haredi community adherence was assessed using [76] ([76]) two-item measure on 7-point scales. The first item measured Halakhic observance, asking participants to position themselves on a continuum from very lenient (1) to very strict (7): “In the Haredi community, there are varying levels of Halacha observance, from very lenient to very strict. Where do you place yourself on this continuum?” The second item assessed adherence to Haredi norms, ranging from low adherence (1) to high adherence (7): “The Haredi community varies from very modern to very strict (e.g., internet use and reading only Haredi publications). Where do you place yourself on this continuum?” The average of these items formed the conservatism score, with higher scores indicating stronger conservatism.

## 3. Results

Analysis of individualism-collectivism patterns in the Haredi community revealed consistent differences across multiple dimensions.

Figure 1 presents the comparisons between balanced and vertical dimensions of collectivism and individualism. Participants demonstrated significantly higher levels of collectivism compared to individualism across both dimensions. The difference was particularly pronounced for the balanced dimension (collectivism: M = 4.22, SD = 0.89; individualism: M = 2.80, SD = 1.17; *p* < 0.001, d = 0.94), with a similarly significant but smaller difference in the vertical dimension (collectivism: M = 4.19, SD = 0.99; individualism: M = 3.34, SD = 1.21; *p* < 0.001, d = 0.49).

Analysis of helping behavior revealed consistent in-group preferences across all three domains examined (Figure 2). The magnitude of this preference varied substantially by type of assistance. Financial help showed the strongest in-group bias (in-group: M = 4.54, SD = 1.36; out-group: M = 2.77, SD = 1.64; *p* < 0.001, d = −0.93), followed by social assistance (in-group: M = 4.18, SD = 1.58; out-group: M = 3.11, SD = 1.74; *p* < 0.001, d = −0.52). Physical help, involving bone marrow donation, demonstrated the smallest, though still significant, in-group preference (in-group: M = 4.89, SD = 1.50; out-group: M = 4.65, SD = 1.67; *p* < 0.001, d = −0.26).

Correlation analysis revealed several meaningful patterns in the relationships between collectivism dimensions and helping behaviors (Table 1). The strongest associations emerged in the consistency of helping patterns across group boundaries. Physical assistance showed a remarkably high correlation between in-group and out-group help (r = 0.83, *p* < 0.001), followed by moderate correlation in social support (r = 0.52, *p* < 0.001).

Within in-group helping behaviors, social and financial assistance demonstrated substantial interconnection (r = 0.52, *p* < 0.001), suggesting a general tendency toward multiple forms of support. The two collectivism dimensions—balanced and vertical—showed moderate correlation (r = 0.49, *p* < 0.001), indicating overlapping but distinct aspects of collectivist orientation.

Collectivist values demonstrated consistent, though modest, associations with helping behaviors. Balanced collectivism correlated more strongly with social support (r = 0.32, *p* < 0.001) than with financial assistance (r = 0.24, *p* < 0.001), while vertical collectivism showed similar patterns (r = 0.27 and r = 0.29, respectively, *p* < 0.001).

To examine the predictors of helping behavior, we conducted a series of linear regression analyses for both in-group and out-group assistance across three domains: social, financial, and physical support (Table 2).

Regression analyses revealed distinct patterns in how cultural orientations and personal characteristics predict helping behaviors across group boundaries. Analysis of physical help demonstrated the highest consistency between in-group and out-group assistance, with in-group physical helping strongly predicting out-group assistance (β = 0.84, *p* < 0.001), explaining 73.5% of the variance. This suggests that when it comes to physical assistance, helping tendencies transcend group boundaries.

Financial assistance showed more complex patterns influenced by both demographic and cultural factors. Out-group financial help was primarily predicted by out-group social help (β = 0.44, *p* < 0.001) and physical help (β = 0.31, *p* < 0.001). For in-group financial assistance, age emerged as a negative predictor (β = −0.10, *p* = 0.001), while balanced collectivism showed a positive association (β = 0.14, *p* < 0.001). Notably, vertical collectivism demonstrated a negative relationship with out-group financial help (β = −0.10, *p* = 0.005), suggesting that hierarchical collective values may limit cross-boundary financial assistance.

For social assistance, cultural orientations played a significant role. The hardiness scale emerged as a significant negative predictor of out-group help (β = −0.17, *p* < 0.001), while vertical collectivism positively predicted in-group social assistance (β = 0.11, *p* = 0.002). Vertical individualism showed a negative relationship with out-group social help (β = −0.07, *p* = 0.041), indicating that hierarchical individualistic values may inhibit cross-boundary social assistance.

The differential effects of collectivism and individualism dimensions reveal that balanced collectivism promotes helping behavior across group boundaries, as evidenced by its positive association with both in-group and out-group financial help (β = 0.14, *p* < 0.001; β = 0.08, *p* = 0.026, respectively). In contrast, vertical orientations, whether collectivistic or individualistic, tend to reinforce in-group preferences while potentially limiting out-group assistance.

## 4. Discussion

This study provides the first empirical examination of collectivistic orientation in the Ultra-Orthodox community using the HVIC (Horizontal–Vertical Individualism–Collectivism) framework. The findings strongly validate previous qualitative observations regarding the community’s collectivistic nature. Participants demonstrated significantly higher scores on both balanced and vertical collectivism compared to individualistic measures, with particularly strong effects in the balanced dimension. The community’s hierarchical structure emerged as a secondary characteristic, evidenced by elevated vertical collectivism scores relative to horizontal measures. This hierarchical pattern aligns with traditional descriptions of the community’s structured social framework ([6]) and corresponds with research showing that traditional societies often exhibit vertical collectivistic tendencies ([113]).

While our study did not include a comparative sample, the HVIC scale employed here has been widely used in cross-cultural studies ([105]; [13]), allowing tentative comparison of average scores. The relatively high mean scores found among our Haredi sample on both horizontal and vertical collectivism suggest a pronounced collectivist orientation compared to previously reported values in Western populations.

Building on this collectivist foundation, the study also explored the relationship between the community’s identity as a “community of kindness’ (i.e., prosocial behavior) and its various underlying characteristics, leading to several significant insights. Within the community, the Hardiness scale was positively associated with acts of prosocial behavior, as expected. Similarly, both scales of collectivism demonstrated a positive influence on prosocial behavior directed toward individuals within the community. However, the patterns shifted notably when examining acts of prosocial behavior extended to individuals outside the community.

The analysis revealed that helping behaviors directed outside the group were inversely related to the Hardiness scale. Individuals with stronger conservative tendencies were less likely to provide help to those outside their immediate community. In contrast, balanced collectivism and balanced individualism were identified as significant predictors of helping behaviors beyond the group, suggesting that a balanced orientation, which integrates both individual and collective priorities, promotes prosocial behavior across social boundaries. Although our findings empirically validate the Haredi community’s strong collectivistic orientation, they also reveal that intrinsic motivation exerts a more substantial influence on out-group helping behavior than communal norms. This underscores the primacy of internal prosocial dispositions over culturally embedded collective expectations in driving altruism beyond group boundaries.

These findings reveal a complex interplay between the collectivistic orientation of the Haredi community and individual motivation to help others. Contrary to dichotomous views that frame individualism and collectivism as mutually exclusive, some scholars suggest that all cultures contain internal heterogeneity in value orientations, allowing for the coexistence of multiple motivational types within the same cultural context ([29]). In line with this perspective, our results indicate that the most influential factor driving out-group helping behavior was not communal norms, but rather individuals’ inherent prosocial disposition. This suggests that helping behaviors are shaped by stable internal traits as much as by cultural expectations. Particularly salient is the role of intrinsically motivated individuals as “internal change agents” who extend the boundaries of communal care and foster connections with broader society. This dynamic functions as a mechanism through which traditional communities preserve their identity while adapting to shifting social realities. A similar pattern appears in [3] ([3]) study of Amish-Mennonite communities, where individual actors, driven by deep religious and familial values, engaged in inter-country adoption—thereby subtly challenging their group’s ethnic and cultural homogeneity. A particularly notable finding emerged in the analysis of different types of assistance, revealing distinct patterns in helping behaviors. While physical help (such as organ donation) showed moderate in-group preferences, financial assistance demonstrated the strongest disparity between in-group and out-group helping. This pattern can be understood through both religious and socioeconomic lenses. The relatively high willingness to provide physical help across group boundaries aligns with fundamental Jewish religious principles, particularly the Talmudic dictum that “whoever saves a life, it is considered as if he saved an entire world” (Sanhedrin 37a). This religious imperative, which emphasizes the supreme value of human life regardless of group affiliation, appears to transcend typical in-group/out-group distinctions. This interpretation is supported by recent findings showing that religious and Ultra-Orthodox individuals in Israel lead in organ donation rates ([11]; [23]), and that they tend to show relative indifference to the recipient’s level of religious observance ([12]), suggesting that when life-saving assistance is involved, religious values may override usual group preferences.

However, this apparent transcendence of group boundaries in organ donation calls for a more nuanced reading. While the recipients in these cases were secular Jews, they remain part of the broader Jewish collective. Studies on organ donation indicate that although Ultra-Orthodox individuals do not distinguish between levels of religious observance, their helping behaviors are typically confined to Jewish recipients ([11], [12]). From this perspective, such acts may be better understood as expressions of internal communal responsibility rather than universal altruism. A key theological framing here is the rabbinic dictum Kol Yisrael arevim zeh la-zeh “All Jews are responsible for one another”, which appears in the Babylonian Talmud (Sanhedrin 27b) and has come to signify a foundational principle of mutual obligation within the Jewish collective ([92]). Even when extended across lines of religious observance, these acts of assistance may still reflect a bounded moral orientation. Compared to physical forms of assistance such as organ donation—which, while generally directed toward Jewish recipients, may occasionally extend beyond one’s immediate religious subgroup—financial helping behaviors exhibit a more marked in-group focus. This difference may be partly explained by distinctions in halakhic obligation: life-saving acts (pikuach nefesh) are considered a binding religious duty that overrides almost all other commandments, potentially encouraging universal assistance when lives are at stake ([8]). In contrast, financial assistance is governed by the principle that “the poor of one’s own town take precedence” (Babylonian Talmud, Bava Metzia 71a), which explicitly prioritizes helping members of one’s own community ([87]). Thus, while religious norms may support broad moral concern in cases of existential threat, they concurrently establish hierarchies of obligation that favor internal redistribution in non-emergency contexts.

Beyond religious motivations, the pronounced in-group preference in financial helping behaviors may also be shaped by the socioeconomic context of the Ultra-Orthodox community. As one of Israel’s economically disadvantaged populations ([59]; [76]), community members may view their limited financial resources as primarily reserved for addressing internal needs. This economic reality, combined with the community’s strong internal support systems, creates a situation where financial assistance remains predominantly within group boundaries, even as other forms of help extend more readily to out-group members.

The pronounced in-group preference in financial assistance observed in this study can be meaningfully interpreted through the principle of subsidiarity—a normative framework asserting that responsibilities should be handled at the most immediate and appropriate level ([28]; [104]). Though rooted in Catholic social thought, subsidiarity provides a useful lens for understanding the Haredi community’s internal logic of resource allocation. It holds that higher-level entities should intervene only when lower-level units cannot effectively meet needs ([31]), thus affirming communal autonomy and localized responsibility. The strong in-group preference in financial aid—especially within an economically challenged community—may reflect the view that such resources are best managed within close-knit networks where trust and knowledge of needs are highest. [46] ([46]) describes subsidiarity as not merely a check on authority but a moral orientation toward proximate decision-making. This reframes in-group preference not as insularity, but as strategic stewardship. The relatively greater openness to physical helping across group lines suggests that subsidiarity operates unevenly across domains, with life-saving acts transcending the typical boundaries of communal prioritization.

While our study focuses on the Ultra-Orthodox community in Israel, the pronounced preference for ingroup financial assistance observed here aligns with patterns documented in other economically disadvantaged collectivist religious groups. Muslim immigrants in the Netherlands, for instance, are more likely to donate to Islamic charities than to secular causes ([17]). In South Asia and North Africa, [36] ([36]) show that the vast majority of zakat payments are made directly to people within givers’ own neighborhoods, kinship networks, and extended families, even where formal collection systems exist. Similarly, in India, donations decrease significantly when recipients are perceived as belonging to lower castes, even among low-income donors ([22]). These findings align with broader evidence linking vertical collectivism with selective prosociality and strong ingroup loyalty, especially under resource scarcity ([33]).

Prosocial behavior plays a crucial role in societal cohesion and sustainability. While previous research has not definitively established the relationship between collectivism and prosocial tendencies, the present study contributes to our understanding of how power dynamics and hierarchical structures influence helping behavior. Our findings suggest that out-group assistance, representing a more purely altruistic form of prosocial behavior, is influenced by power orientation. Specifically, individuals with horizontal orientations, who view others as equals, demonstrate greater willingness to extend help beyond group boundaries compared to those with vertical orientations.

### Study Limitations and Directions for Future Research

The current study includes certain methodological considerations. One aspect worth noting is that some scales exhibited relatively low alpha scores, which may reflect the unique characteristics of this population rather than inherent measurement limitations. While internationally standardized measures provided valuable insights into collectivism within the Ultra-Orthodox community, further refinements may enhance their ability to fully capture its nuanced social and cultural dynamics. Nonetheless, the findings strongly support the community’s collectivistic orientation and highlight the relevance of adapting measurement tools to align more closely with its distinct characteristics ([86]). Second, as this study is correlational, it does not establish whether internal prosocial tendencies drive out-group prosocial behavior or if engaging in out-group helping reinforces internal prosocial dispositions. Experimental research could provide further insights to clarify the directionality of this relationship.

Beyond these methodological limitations, it is important to consider the broader implications of these findings for understanding the interactions between the Haredi community and Israeli society at large. A significant insight from our research is the inherent tension between the community’s dominant collectivist structure and the individualistic traits that emerged in our findings. This tension reveals how individuals with stronger inherent prosocial dispositions function as natural internal change agents, gradually shifting community boundaries through their personal characteristics rather than through external pressures. This organic process of community evolution through individual agency represents a critical mechanism through which traditional collectivist societies can adapt while maintaining their core identity.

The tension between vertical and horizontal collectivism identified in our research reflects not only internal community dynamics but also the challenges and opportunities that exist in the encounter between different worldviews in the Israeli context. The Haredi community, with its distinct collectivist characteristics, exists within a state that combines both individualistic and collectivist elements. Understanding the factors that promote cross-boundary assistance may help develop policies and intervention programs that strengthen general solidarity in Israeli society while respecting and recognizing the unique characteristics of the Haredi community.

Future research could deepen our understanding in several complementary directions. First, a more thorough examination of the dynamics between horizontal and vertical collectivism, as reflected in different subgroups within the Haredi community, especially in light of the distinction between halakhic conservatism and sociological conservatism. Second, longitudinal research would allow tracking changes in helping patterns and collectivist characteristics over time, particularly among young Haredim who are exposed to more external influences through integration into the workforce and higher education. Third, comparative research between Haredi communities in Israel and the diaspora could provide insights into the influence of broader cultural and social contexts on the relationship between collectivism and prosocial behavior. Finally, developing measurement tools specifically adapted to the Haredi population could enable a more precise understanding of the cultural and psychological structures unique to this community, and enrich our understanding of the diverse expressions of collectivism in different societies.

## Figures and Tables

**Figure 1 behavsci-15-00520-f001:**
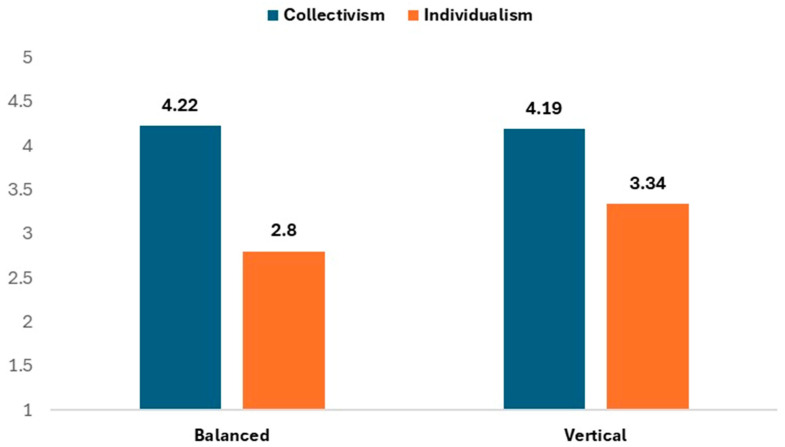
Present’s descriptive statistics and paired *t*-tests for the differences in balanced versus vertical collectivism and individualism.

**Figure 2 behavsci-15-00520-f002:**
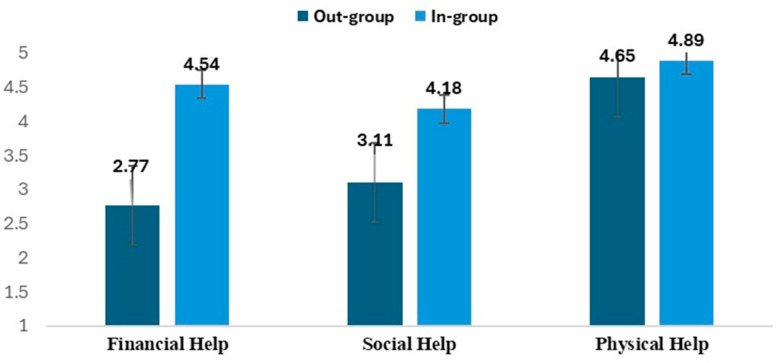
Present’s descriptive statistics and paired *t*-tests for the differences in assistance provided to in-group versus out-group members across social, financial, and physical domains.

**Table 1 behavsci-15-00520-t001:** Means, standard deviations, and correlation matrix for age and conservativeness, collectivism, individualism, help to the internal group, help to the external group.

Measure	Mean	SD	1	2	3	4	5	6	7	8	9	10	11	12	13
1. gender	1.36	0.48	(-)												
2. age	30.98	11.82	−0.03	(-)											
3. hardiness scale	4.37	1.42	0.09 *	0.03	(-)										
4. balanced collectivism	4.22	0.89	−0.04	0.02	0.18 ***	(-)									
5. balanced individualism	2.8	1.17	0.08 *	0.15 ***	0.13 ***	−0.08 *	(-)								
6. vertical collectivism	4.19	0.99	−0.07	0.03	0.21 ***	0.49 ***	−0.03	(-)							
7. vertical individualism	3.34	1.21	0.19 ***	0.05	−0.04	−0.26 ***	0.26 ***	−0.24 ***	(-)						
8. help out group social	3.11	1.74	−0.10 **	−0.01	−0.25 ***	0.08 *	−0.19 ***	0.06	−0.11 **	(-)					
9. help out group money	2.77	1.64	−0.06	−0.02	−0.20 ***	0.11 **	−0.15 ***	−0.02	−0.02	0.52 ***	(-)				
10. help out group body	4.65	1.67	0.10 **	−0.08 *	−0.02	0.20 ***	−0.12 **	0.13 ***	−0.03	0.23 ***	0.28 ***	(-)			
11. help in social	4.18	1.58	0.01	0.04	0.15 ***	0.24 ***	−0.06	0.29 ***	−0.12 **	0.23 ***	0.11 **	0.17 ***	(-)		
12. help in money	4.54	1.36	0.02	−0.0 9*	0.16 ***	0.32 ***	−0.06	0.27 ***	−0.12 **	0.11 **	0.20 ***	0.24 ***	0.52 ***	(-)	
13. help in body	4.89	1.5	0.07	−0.0 8*	0.09 *	0.24 ***	−0.09 *	0.18 ***	−0.07	0.12 **	0.15 ***	0.83 ***	0.29 ***	0.36 ***	(-)

Note. N = 702. * *p* < 0.05. ** *p* < 0.01. *** *p* < 0.001.

**Table 2 behavsci-15-00520-t002:** Linear regression analysis predicting in-group and out-group helping behaviors across social, financial, and physical domains.

Predictors	Help Out Group Social		Help Out Group Money		Help Out Group Body		Help in Social		Help in Money		Help in Body	
	Std. Beta	*p*	Std. Beta	*p*	Std. Beta	*p*	Std. Beta	*p*	Std. Beta	*p*	Std. Beta	*p*
(Intercept)	0	<0.001	0	0.067	−0.00	0.327	0	0.157	0	0.003	0	0.001
gender	−0.05	0.145	−0.04	0.195	0.06	0.006	0.03	0.382	0.01	0.672	−0.02	0.287
age	0.01	0.792	0.02	0.608	−0.01	0.642	0.08	0.008	−0.10	0.001	−0.01	0.558
hardiness scale	−0.17	<0.001	−0.07	0.028	−0.03	0.154	0.08	0.014	0.05	0.099	0.03	0.119
balanced individualism	−0.06	0.052	−0.04	0.245	−0.03	0.194	−0.02	0.645	0	0.938	0.01	0.732
balanced collectivism	−0.02	0.517	0.08	0.026	0.03	0.205	0	0.971	0.14	<0.001	0.01	0.676
vertical collectivism	0.05	0.149	−0.10	0.005	0.02	0.4	0.11	0.002	0.06	0.108	−0.01	0.793
vertical individualism	−0.07	0.041	0.05	0.107	0.03	0.132	0	0.971	−0.01	0.713	−0.02	0.305
help group money	0.43	<0.001			0.12	<0.001	−0.07	0.057	0.2	<0.001	−0.08	0.001
help group body	0.21	0.001	0.31	<0.001			−0.18	0.002	−0.16	0.006	0.82	<0.001
help in social	0.25	<0.001	−0.07	0.057	−0.07	0.002			0.4	<0.001	0.1	<0.001
help in money	−0.09	0.018	0.21	<0.001	−0.07	0.006	0.42	<0.001			0.13	<0.001
help in body	−0.15	0.01	−0.20	0.001	0.84	<0.001	0.25	<0.001	0.3	<0.001		
help group social			0.44	<0.001	0.08	0.001	0.25	<0.001	−0.09	0.018	−0.06	0.01
Observations	702		702		702		702		702		702	
R^2^/R^2^ adjusted	0.367/0.356		0.351/0.339		0.740/0.735		0.360/0.349		0.388/0.378		0.744/0.740	

## Data Availability

The data presented in this study are available in OSF at https://osf.io/t3v6y/files/osfstorage (15 February 2025).

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
