# Peer review of "Helping Across Boundaries: Collectivism and Hierarchy in the Ultra-Orthodox Context"

_behavsci, 2025, doi:10.3390/bs15040520_

Round 1
Reviewer 1 Report
Comments and Suggestions for Authors
Personal identities can be an enigma emitting fragments of understanding through time. These fragments of awakening can produce turbulence and become necessary steps towards growth and transformation. In a social setting where culture, rituals, customs, narratives, practices and perceptions have been built up forming a tradition and embracing of the sacred reality of life, the existence of personal identity takes on a communal lens and function. Individuation and growth in the social group are intensively reinforced with careful surveillance, vigilance and commitment. The future is at stake, and families becomes centres of newness as much as tradition. The importance of the rebbe or spiritual leader in Hasidism has a mediating, sacred function to invite holiness and goodness into the community, and hence a sense of direction and purpose. Otherness or altruism can be practices naturally (of lived experience) within the group, and the extent to which it overflows within society will depend on a number of factors such as religious missionary zeal or possessing a spontaneous sense of faith to care for the neighbour. The question remains whether “in group solidarity” can be enunciated as solidarity (and subsidiarity) in the community at large, and hence outside the group.
Developing a spontaneous hospitality towards others outside the group is a challenge not just for conservative and traditionalists groups, but for all societies. Nevertheless, ultra-orthodox groups, whether Jewish or Christian for example, can be secretive, excessive and exclusive. Equality itself speaks of a future world otherwise than privilege and exclusion. To enunciate equality takes on even religious, sacred overtones signifying care for others. As the communal sense of keeping the well-being of others in mind and heart wrestles with the individual’s self-interested desires, the communal structure and practices act as a framework to orient consciousness towards alterity (“an individual’s inherent disposition to help”).
This is a worthy, well-written and researched and fascinating study of the Ultra-Orthodox (Haredi) community, and my comments offer more a conversation to inspire reflection on otherness. The findings will bring greater understanding and respect to the “enclave culture” of the Haredi community, necessary for peaceful co-existence to journey towards the future.
Suggestions to consider:
1. The article may also want to engage the ideas of subsidiarity which is connected to solidarity. Subsidiarity keeps in mind the centrality of the human person. This could be very challenging for the Haredi community. In this way, the subsidiarity of affirming an individual’s rights and autonomy within the community can help to nurture a greater commitment to the Haredi culture, life and faith, and thus help to balance the communal with the self’s freedom to choose and to grow. Such tension can create newness and growth animating the community to find new ways of responding to modernity and the nation’s future and needs. In this way, Israel as Torah can have a greater meaning.
2. How does the “intrinsic” motivation to help others give a sense of newness to the community? By pushing boundaries of care towards others? Through developing a natural boldness to go to the margins beyond the community?
3. Provide a couple more paragraphs of reflection in the concluding section, “Study Limitations and Directions for Future Research” to give greater clarity and import of the paper.
Overall, again, this is a manuscript worthy of publication that can do much to encourage understanding sensitivity, dialogue and respect for the Haredi Community as much as helping the direction and future of society in Israel.
Author Response
Reviewer 1:
Comment: The article may also want to engage the ideas of subsidiarity which is connected to solidarity. Subsidiarity keeps in mind the centrality of the human person. This could be very challenging for the Haredi community.
Response: We thank the reviewer for this insightful suggestion. We were not previously familiar with the concept of subsidiarity, but upon researching it, we found it provides an excellent organizing framework that logically explains the internal resource allocation logic of the Haredi community. We have incorporated this concept into our revised manuscript, highlighting how subsidiarity—the principle that matters should be handled by the smallest, lowest, or least centralized competent authority—helps explain the pronounced in-group preference in financial assistance observed in our study. As can be seen on page 27-28:
The pronounced in-group preference in financial assistance observed in this study can be meaningfully interpreted through the principle of subsidiarity—a normative framework asserting that responsibilities should be handled at the most immediate and appropriate level (Evans & Zimmermann, 2014). Though rooted in Catholic social thought, subsidiarity provides a useful lens for understanding the Haredi community’s internal logic of resource allocation. It holds that higher-level entities should intervene only when lower-level units cannot effectively meet needs (Finnis, 2016), thus affirming communal autonomy and localized responsibility. The strong in-group preference in financial aid—especially within an economically challenged community—may reflect a view that such resources are best managed within close-knit networks where trust and knowledge of needs are highest. Halberstam (2009) describes subsidiarity as not merely a check on authority but a moral orientation toward proximate decision-making. This reframes in-group preference not as insularity, but as strategic stewardship. The relatively greater openness to physical helping across group lines suggests that subsidiarity operates unevenly across domains, with life-saving acts transcending the typical boundaries of communal prioritization.
Comment: How does the "intrinsic" motivation to help others give a sense of newness to the community?
Response: Thank you for this insightful comment highlighting the importance of intrinsic motivation as a driver of community renewal. We have emphasized this key finding throughout our revised manuscript. As can be seen on the following pages:
Page 1- Abstract:
The strongest predictor of out-group assistance was an individual's inherent disposition to help, suggesting that prosocial behavior extends beyond purely communal expectations and positions these individuals as natural agents of community change. This insight offers a perspective on how personal characteristics may contribute to community renewal
Page 25-26 Discussion
Although our findings empirically validate the Haredi community’s strong collectivistic orientation, they also reveal that intrinsic motivation exerts a more substantial influence on out-group helping behavior than communal norms. This underscores the primacy of internal prosocial dispositions over culturally embedded collective expectations in driving altruism beyond group boundaries.
These findings reveal a complex interplay between the collectivistic orientation of the Haredi community and individual motivation to help others. Contrary to dichotomous views that frame individualism and collectivism as mutually exclusive, some scholars suggest that all cultures contain internal heterogeneity in value orientations, allowing for the coexistence of multiple motivational types within the same cultural context (Kamal Fatehi et al., 2020). In line with this perspective, our results indicate that the most influential factor driving out-group helping behavior was not communal norms, but rather individuals’ inherent prosocial disposition. This suggests that helping behaviors are shaped by stable internal traits as much as by cultural expectations. Particularly salient is the role of intrinsically motivated individuals as "internal change agents" who extend the boundaries of communal care and foster connections with broader society. This dynamic functions as a mechanism through which traditional communities preserve their identity while adapting to shifting social realities. A similar pattern appears in Anderson and Anderson’s (2023) study of Amish-Mennonite communities, where individual actors, driven by deep religious and familial values, engaged in inter-country adoption—thereby subtly challenging their group’s ethnic and cultural homogeneity.
Page 29-30 - Study Limitations and Directions for Future Research
Beyond these methodological limitations, it is important to consider the broader implications of these findings for understanding the interactions between the Haredi community and Israeli society at large. A significant insight from our research is the inherent tension between the community's dominant collectivist structure and the individualistic traits that emerged in our findings. This tension reveals how individuals with stronger inherent prosocial dispositions function as natural internal change agents, gradually shifting community boundaries through their personal characteristics rather than through external pressures. This organic process of community evolution through individual agency represents a critical mechanism through which traditional collectivist societies can adapt while maintaining their core identity.
The tension between vertical and horizontal collectivism identified in our research reflects not only internal community dynamics but also the challenges and opportunities that exist in the encounter between different worldviews in the Israeli context. The Haredi community, with its distinct collectivist characteristics, exists within a state that combines both individualistic and collectivist elements. Understanding the factors that promote cross-boundary assistance may help develop policies and intervention programs that strengthen general solidarity in Israeli society while respecting and recognizing the unique characteristics of the Haredi community.
Comment: Provide a couple more paragraphs of reflection in the concluding section, "Study Limitations and Directions for Future Research."
Response: We have substantially expanded the "Study Limitations and Directions for Future Research" , adding reflections on the broader implications of our findings, the tension between different forms of collectivism, and specific avenues for future inquiry. As can be seen on page 29-32
Future research could deepen our understanding in several complementary directions. First, a more thorough examination of the dynamics between horizontal and vertical collectivism as reflected in different subgroups within the Haredi community, especially in light of the distinction between halakhic conservatism and sociological conservatism. Second, longitudinal research would allow tracking changes in helping patterns and collectivist characteristics over time, particularly among young Haredim who are exposed to more external influences through integration into the workforce and higher education. Third, comparative research between Haredi communities in Israel and the diaspora could provide insights into the influence of broader cultural and social contexts on the relationship between collectivism and prosocial behavior. Finally, developing measurement tools specifically adapted to the Haredi population could enable a more precise understanding of the cultural and psychological structures unique to this community, and enrich our understanding of the diverse expressions of collectivism in different societies.
We sincerely thank the reviewer for the thoughtful and constructive feedback
Reviewer 2 Report
Comments and Suggestions for Authors
Review
Helping Across Boundaries: Collectivism and hierarchy in the Ultra-Orthodox Context
I thank the editorial team for the opportunity to read this interesting article on an important topic. This research examines collectivism within the Ultra-Orthodox (Haredi) community using the Horizontal and Vertical Individualism-Collectivism framework, investigating how collectivist and individualist orientations influence prosocial behaviors toward in-group and out-group members. The study contributes to cross-cultural research by revealing the complex interplay between collectivism, power orientation, and helping tendencies, demonstrating that while conservatism predicts in-group assistance, balanced collectivism and individualism facilitate out-group helping behaviors.
That said, I would also like to draw attention to several aspects that might benefit from further consideration in the text:
- Regarding the statement that "The present study applies these theoretical frameworks to Israel's Ultra-Orthodox (Haredi) community, a distinct and understudied population," I would note that this characterization may need refinement. Over the past decade, there has been a growing body of diverse research examining various aspects of Haredi society, including several studies specifically investigating collectivism within this community from different perspectives. The existing literature is more substantial than the statement suggests, and acknowledging these contributions would strengthen the contextual positioning of your research.
- Throughout the theoretical review within each section, there is a reliance on a range of sources, but some of those sources are not up-to-date. In general, it would be worthwhile to include additional, newer sources.
- The paper would benefit from a more nuanced discussion of the relationship between religious values and prosocial behavior. While the authors highlight the Talmudic dictum about saving lives to explain physical helping behaviors across group boundaries, the analysis could be strengthened by exploring how other religious teachings might influence the patterns observed in different types of assistance. This would provide a more comprehensive framework for understanding the interplay between religious identity and prosocial tendencies.
- The discussion of economic factors influencing in-group financial assistance could be enhanced by incorporating comparative data from other collectivist communities with similar socioeconomic challenges. This would help clarify whether the observed patterns are unique to the Ultra-Orthodox community or represent broader tendencies in economically disadvantaged collectivist groups, thereby strengthening the generalizability of the findings.
- Future studies should explore the developmental trajectory of prosocial orientation within the Ultra-Orthodox community, particularly how balanced collectivism emerges across different life stages. Longitudinal research examining how prosocial behaviors toward in-group and out-group members change throughout an individual's life cycle would provide valuable insights into the formation and evolution of these tendencies within collectivist religious communities.
Author Response
Reviewer 2:
Comment: Over the past decade, there has been a growing body of diverse research examining various aspects of Haredi society, including several studies specifically investigating collectivism within this community from different perspectives.
Response: Thank you for this valuable observation. We have revised our characterization in the introduction (p. 1) and expanded our literature review of research on the Ultra-Orthodox society (pp. 12-16). We now acknowledge the substantial body of work examining various aspects of Haredi life while clarifying our specific contribution regarding empirical measurement of collectivistic tendencies using standardized measures. You can see the changes in the following sources:
Page 11:
In a study of national civic service, Malchi and Ben Porat (2018) found that most young Haredi men preferred to volunteer within their own community, citing a sense of security and belonging associated with bonding social capital. However, a smaller number opted for extra-community service, motivated by bridging social capital and a desire for broader integration into Israeli society.
Page 13:
In recent decades, however, the ultra-Orthodox community has undergone significant social and ideological transformations. Its social margins have expanded, with new subgroups and ideological currents emerging (Kook & Harel-Shalev, 2020; Haller, 2024; Abu-Kaf, at el, 2023). Increasing numbers of Haredi individuals have entered higher education, the workforce, and public life more broadly (Kalagy, 2018; 2020), while a nascent middle-class lifestyle has begun to develop within parts of the community, reflecting greater openness to modernization and societal integration (Suzin, 2025).
Comment: Throughout the theoretical review within each section, there is a reliance on a range of sources, but some of those sources are not up-to-date.
Response: Thank you for this comment. We have substantially updated our literature review throughout the manuscript to include more recent sources, providing a more current perspective on the relevant topics. We have added 10 new sources from recent years, such as:
Booysen, F., Guvuriro, S., & Campher, C. (2021). Horizontal and vertical individualism and collectivism and preferences for altruism: A social discounting study. Personality and Individual Differences, 178, 110856. https://doi.org/10.1016/j.paid.2021.110856
Čavojová, V., Adamus, M., & Ballová Mikušková, E. (2022). You before me: How vertical collectivism and feelings of threat predicted more socially desirable behaviour during COVID-19 pandemic. Current Psychology, 43(9), 8303– 8314. https://doi.org/10.1007/s12144-022-03003-3
Cheng, A. W., Rizkallah, S., & Narizhnaya, M. (2020). Individualism vs. collectivism. The Wiley Encyclopedia of Personality and Individual Differences: Clinical, Applied, and Cross‐Cultural Research, 287-297. https://doi.org/10.1002/9781118970843.ch313
Fatehi, K., Priestley, J. L., & Taasoobshirazi, G. (2020). The expanded view of individualism and collectivism: One, two, or four dimensions? International Journal of Cross Cultural Management, 20(1), 7–24. https://doi.org/10.1177/1470595820913077
Jiao, J., & Zhao, J. (2023). Individualism, collectivism, and allocation behavior: Evidence from the ultimatum game and dictator game. Behavioral Sciences, 13(2), 169. https://doi.org/10.3390/bs13020169
Lykes, V. A., & Kemmelmeier, M. (2014). What predicts loneliness? Cultural difference between individualistic and collectivistic societies in Europe. Journal of Cross-Cultural Psychology, 45(3), 468-490. https://doi.org/10.1177/0022022113509881
Vignoles, V. L., Owe, E., Becker, M., Smith, P. B., Easterbrook, M. J., Brown, R., & Bond, M. H. (2016). Beyond the “East–West” dichotomy: Global variation in cultural models of selfhood. Journal of Experimental Psychology: General, 145(8), 966–1000. https://doi.org/10.1037/xge0000175
Comment: The paper would benefit from a more nuanced discussion of the relationship between religious values and prosocial behavior.
Response: Thank you for this comment, We have added a new section titled "Religiosity and Group-Boundaries in Prosocial Behavior" (pp. 10-12) that reviews empirical literature on this topic. We have also enhanced our discussion of how religious principles guide different types of helping behaviors as can be seen.
Page 10: Religiosity and Group-Boundaries in Prosocial Behavior
The relationship between religiosity and prosocial behavior has been extensively explored, particularly with regard to in-group versus out-group helping. A recent meta-analysis found a small but robust positive association between religiosity and prosociality, with stronger effects for self-reported behavior than for behavioral measures. Notably, religious individuals were more inclined to help in-group members, although this tendency was moderate in size (Kelly et al., 2024). This pattern is supported by studies reporting selective prosociality among religious populations (Blogowska & Saroglou, 2011; Różycka-Tran, 2017; Speed & Brewster, 2021). These studies indicate a tendency to favor co-religionists and exhibit reduced support or more negative attitudes toward out-groups such as atheists or members of other religions.
In contrast, several studies suggest that religiosity can promote generalized prosociality, particularly under certain cognitive or contextual conditions. Experimental findings show that religious priming enhances generosity toward both in-group and out-group members (Bryan et al., 2016; Pasek et al., 2023). Similar effects were observed among Muslim minorities, where priming increased donations primarily toward non-Muslim recipients (Morton et al., 2018). A field experiment in China found that Christians and Buddhists showed no clear in-group preference, whereas Muslim participants favored their own group (Xia et al., 2021). Additional support for the link between religiosity and broader altruistic orientation comes from a study among Jewish populations in Israel, which found that higher levels of religiosity were positively associated with general altruistic tendencies (Dopelt et al., 2022). A systematic review identified mechanisms through which religiosity shapes giving to both in-group and out-group recipients, highlighting cross-religious variation (Yasin et al., 2020).
Taken together, the literature suggests a dual dynamic: while religiosity often enhances prosocial behavior toward the in-group, it may also motivate helping across group boundaries when moral reflection, minority status, or religious mission are salient. This distinction is particularly relevant in collectivist religious communities, such as the Ultra-Orthodox population studied here.
Comment: The discussion of economic factors influencing in-group financial assistance could be enhanced by incorporating comparative data from other collectivist communities with similar socioeconomic challenges.
Response: Thank you for this comment. We have added comparative data from other economically disadvantaged collectivist religious groups to strengthen our argument that the observed patterns reflect broader tendencies in economically challenged collectivist communities (pp. 29-30).
While our study focuses on the Ultra-Orthodox community in Israel, the pronounced preference for ingroup financial assistance observed here aligns with patterns documented in other economically disadvantaged collectivist religious groups. Muslim immigrants in the Netherlands, for instance, are more likely to donate to Islamic charities than to secular causes (Carabain & Bekkers, 2012). In South Asia and North Africa, Gallien et al. (2024) show that the vast majority of zakat payments are made directly to people within givers’ own neighborhoods, kinship networks, and extended families, even where formal collection systems exist. Similarly, in India, donations decrease significantly when recipients are perceived as belonging to lower castes, even among low-income donors (Deshpande & Spears, 2016). These findings align with broader evidence linking vertical collectivism with selective prosociality and strong ingroup loyalty, especially under resource scarcity (Fischer & Derham, 2016).
Comment: Future studies should explore the developmental trajectory of prosocial orientation within the Ultra-Orthodox community.
Response: Thank you for this comment. We have incorporated this suggestion into our paper as can be seen in pages 26:
However, this apparent transcendence of group boundaries in organ donation calls for a more nuanced reading. While the recipients in these cases were secular Jews, they remain part of the broader Jewish collective. Studies on organ donation indicate that although Ultra-Orthodox individuals do not distinguish between levels of religious observance, their helping behaviors are typically confined to Jewish recipients (Boaz, 2022, 2023). From this perspective, such acts may be better understood as expressions of internal communal responsibility rather than universal altruism. A key theological framing here is the rabbinic dictum Kol Yisrael arevim zeh la-zeh “All Jews are responsible for one another”, which appears in the Babylonian Talmud (Sanhedrin 27b) and has come to signify a foundational principle of mutual obligation within the Jewish collective (Rudman, 2009). Even when extended across lines of religious observance, these acts of assistance may still reflect a bounded moral orientation. Compared to physical forms of assistance such as organ donation—which, while generally directed toward Jewish recipients, may occasionally extend beyond one’s immediate religious subgroup—financial helping behaviors exhibit a more marked in-group focus. This difference may be partly explained by distinctions in halakhic obligation: life-saving acts (pikuach nefesh) are considered a binding religious duty that overrides almost all other commandments, potentially encouraging universal assistance when lives are at stake (Bleich, 2002). In contrast, financial assistance is governed by the principle that “the poor of one’s own town take precedence” (Babylonian Talmud, Bava Metzia 71a), which explicitly prioritizes helping members of one’s own community (Rapfogel, 2004). Thus, while religious norms may support broad moral concern in cases of existential threat, they concurrently establish hierarchies of obligation that favor internal redistribution in non-emergency contexts.
Beyond religious motivations, the pronounced in-group preference in financial helping behaviors may also be shaped by the socioeconomic context of the Ultra-Orthodox community. As one of Israel's economically disadvantaged populations (Kasir & Tzachor-Shai, 2017; Malovicki-Yaffe, et al., 2018),
And pages 29-30
Future research could deepen our understanding in several complementary directions. First, a more thorough examination of the dynamics between horizontal and vertical collectivism as reflected in different subgroups within the Haredi community, especially in light of the distinction between halakhic conservatism and sociological conservatism. Second, longitudinal research would allow tracking changes in helping patterns and collectivist characteristics over time, particularly among young Haredim who are exposed to more external influences through integration into the workforce and higher education. Third, comparative research between Haredi communities in Israel and the diaspora could provide insights into the influence of broader cultural and social contexts on the relationship between collectivism and prosocial behavior. Finally, developing measurement tools specifically adapted to the Haredi population could enable a more precise understanding of the cultural and psychological structures unique to this community, and enrich our understanding of the diverse expressions of collectivism in different societies.
"Directions for Future Research" section (p. 32), recommending longitudinal studies to track changes in helping patterns and collectivist characteristics over time.
We thank you again for the opportunity to revise our manuscript and believe that these changes have significantly improved its quality and contribution to the field.
Sincerely,
The Authors